# Vitamin D Doses from Solar Ultraviolet and Dietary Intakes in Patients with Depression: Results of a Case-Control Study

**DOI:** 10.3390/nu12092587

**Published:** 2020-08-26

**Authors:** Haitham Jahrami, Nicola Luigi Bragazzi, William Burgess Grant, Hala Shafeeq Mohamed AlFarra, Wafa Shafeeq Mohamed AlFara, Shahla Mashalla, Zahra Saif

**Affiliations:** 1Ministry of Health, Manama, Bahrain, P.O. Box 12 Manama, Bahrain; hfara@health.gov.bh (H.S.M.A.); wshafiq@health.gov.bh (W.S.M.A.); smaki@health.gov.bh (S.M.); zsaif@health.gov.bh (Z.S.); 2College of Medicine and Medical Sciences, Arabian Gulf University, P.O. Box 26671 Manama, Bahrain; 3Laboratory for Industrial and Applied Mathematics (LIAM), Department of Mathematics and Statistics, York University, ON M3J 1P3, Canada; robertobragazzi@gmail.com; 4Sunlight, Nutrition, and Health Research Center, P.O. Box 641603, San Francisco, CA 94164-1603, USA; wbgrant@infionline.net

**Keywords:** 25OHD, mood disorders, UVB, vitamin D analogs, vitamin D supplementation

## Abstract

The purpose of this study to estimate cumulative vitamin D doses from solar ultraviolet and dietary intakes in patients with depression and compare it to healthy controls. Using a case-control research design, a sample of 96 patients with depression were age- and sex-matched with 96 healthy controls. Dietary vitamin D dose was estimated from diet analysis. Vitamin D-weighted ultraviolet solar doses were estimated from action spectrum conversion factors and geometric conversion factors accounting for the skin type, the fraction of body exposed, and age factor. Patients with depression had a lower dose of vitamin D (IU) per day with 234, 153, and 81 per day from all sources, sunlight exposure, and dietary intake, respectively. Controls had a higher intake of vitamin D (IU) per day with 357, 270, and 87 per day from all sources, sunlight exposure, and dietary intake, respectively. Only 19% and 30% met the minimum daily recommended dose of ≥400 IU per day for cases and controls, respectively. The sensitivity, specificity, percentage correctly classified and receiver operating characteristic (ROC) Area for the estimated vitamin D against serum vitamin D as reference were 100%, 79%, 80%, and 89%. Physical activity level was the only predictor of daily vitamin D dose. Vitamin D doses are lower than the recommended dose of ≥400 IU (10 mcg) per day for both cases with depression and healthy controls, being much lower in the former.

## 1. Introduction

Depression is a universal mental illness that affects a large proportion of any community [1]. A recent meta-analysis showed that the estimated point-, 12-months, and lifetime-prevalence rates of depression are 12.9%, 7.2%, and 10.8%, respectively [1]. The illness affects 350 million persons worldwide and is considered a leading cause of disability [2]. The first line of treatment for major depression is pharmacotherapy [3]. Recent network meta-analyses show that drug pharmacotherapy demonstrates minimal difference from placebo [4,5]. Other modalities such as electroconvulsive therapy and psychotherapy also showed similar results compared to sham procedures [6,7,8]. Dietary and lifestyle approaches hold potential as a novel intervention for the management of symptoms of depression [9]. They can be used in support of pharmacotherapy for severe cases. Therefore, understanding the specific components of dietary and lifestyle interventions that improve mental health are needed. 

The association between depression and the status of vitamin D from lack of sun exposure is well established and was first described two thousand years ago [10]. Results from epidemiological studies shows that vitamin D deficiency is associated with an 8%–14% increase in depression [11] and a 50% increase in suicide [12]. In the past 10 years, an increasing body of literature has linked vitamin D to the pathophysiology of depression [13]. This comes from three lines of evidence; first, the presence of vitamin D receptors in various parts of the cortex and limbic system [14]; second, the important modulatory role that vitamin D plays in regulating immunoinflammatory pathways that are relevant to the pathophysiology of depression [15,16]; third, lower serum vitamin D levels in depressed patients compared to controls [13,17,18]. The reasons for the difference in serum vitamin D between cases with depression and controls remained unclear. 

Vitamin D deficiency for patients with depression as well as a healthy population has become an important community health concern. Previous research focused on either laboratory approaches of measuring vitamin D serum 25-hydroxyvitamin D 25(OH)D [19,20], or focused on establishing an association between diet or dietary supplements and vitamin D [21]. No previous work was established to cover the estimation of solar ultraviolet (UV) doses of patients with depression and vitamin D3 production. Accordingly, the current study was designed to estimate how much vitamin D3 is acquired from diet and produced from everyday outdoor ultraviolet type B doses in Bahrain (26 °N) for cases with depression in comparison to age- and sex-matched controls. We hypothesize that healthy controls acquire higher daily vitamin D3 doses from both dietary and sunlight exposure compared to cases. We also hypothesize that the severity of depressive symptoms is associated with the level of vitamin D3 acquired. 

## 2. Materials and Methods

### 2.1. Study Design and Setting

The current research utilized the guidelines of the strengthening the reporting of observational studies in epidemiology (STROBE) statement [22]. The study took place between March and December 2019. 

Cases with depression were recruited from the outpatient clinics of the general adult psychiatry services of the Psychiatric Hospital, Ministry of Health, Manama, Kingdom of Bahrain. The Psychiatric Hospital, Bahrain, is the national center for mental illness in Bahrain. The hospital registry shows that there are about 750 cases with depression only without another psychiatric morbidity. Controls were recruited from local health centers during regular non-emergency visits, and routine/investigation visits. The local health centers are the primary healthcare clinics belonging to the Ministry of Health, Bahrain. 

### 2.2. Participants

Cases—We included cases with depression (major depressive disorder, single episode, unspecified depressive disorder). Diagnosis was made using the International Statistical Classification of Diseases, 10th Revision. We included adults aged between 20–60 years who were diagnosed in the past six months or more using the ICD-10 criteria. We excluded: women who are pregnant or lactating; the coexistence of any other psychiatric disorder, e.g., eating disorder, generalized anxiety disorder, etc.; or those who were dieting, taking dietary supplements, or enrolled in lifestyle experimental studies. 

Controls—We included controls, defined as individuals free from a known history of mental illness including depression. Controls were achieved by matching each case with depression with a person from the local care centers. Age match was on the basis of year of birth. We excluded: women who are pregnant or lactating, positive history of psychiatric disorder, those who were dieting, taking dietary supplements, or enrolled in lifestyle experimental studies. 

### 2.3. Sample Size and Sampling Techniques

Using a matched case-control design, we estimated the sample needed for our research to be 75 patients and 75 controls. Sample size calculations are based on a z test, with a 1:1 ratio design assuming the difference in vitamin D3 intake by 33% based on previous research [23]. The sample size was estimated for the two-sided test with error probabilities of alpha = 0.05 and 80% power (beta = 0.20). To further increase the statistical power, we aimed to include 95–100 patients in each group. 

Probability sampling techniques were used for recruiting cases and controls. The sample of depression cases (*n* = 96) was selected using a simple random sampling technique from the case registry. Similarly, controls (*n* = 96) were selected using simple random sampling after matching. 

### 2.4. Data Collection Procedure

Data were collected using structured forms and included sociodemographic and anthropometric variables, medical history, and comprehensive lifestyle assessment. The anthropometric measurements included weight, height, and body composition analysis. Weight was measured using electronic scales with rod height attachment. During measurements, individuals were advised to stand straight, without footwear, and keep on only light clothes. Body composition analysis (BCA) was completed using a bioelectrical impedance system (The InBody 230 model: MW160, Seoul/Korea). BCA involved fat mass, and body fat percentage. Body mass index (BMI) (kg/m2) was classified corresponding to the World Health Organization (WHO) categories of underweight (<18.5), normal (18.5–24.9), overweight (25.0–29.9), or obese (≥30) [24]. 

The electronic medical record was accessed to obtain data available in the past six months from the interview on serum vitamin D, and no special request was made to collect a new blood sample. Vitamin D was analyzed as 25(OH)D using a chemiluminescent immunoassay in our study. This method (in Ministry of Health, Bahrain laboratories) has a correlation coefficient with the high performance liquid chromatography assay of 0.92. 

For cases with depression, the Beck Depression Inventory-II (BDI-II) was used to assess the severity of symptoms. The BDI-II is a sum score of all 21 items of the scale; each item is evaluated on a 4 points (0–3) Likert scale [25]. The following algorithm has been used to interpret the BDI-II: minimal depression = 0–13, mild depression = 14–19, moderate depression = 20–28, and severe depression = 29–63. We used the validated Arabic version of the BDI-II in our study [26]. 

A quantitative food frequency questionnaire (covering 102 foods distributed on 38 items/groups) was used [27]. Participants were requested to report the frequency of consuming a standard serving of a specific food item in six categories (1 time/day, ≥2 times/day, 1–2 times/week, 3–6 times/week, 1–3 times/month, rarely, or never). Special attention was given to vitamin D rich food including fatty fish, liver, meat, cheese, eggs, dairy products, and foods fortified with vitamin D such as juices and cereals [28]. The responses on the frequent consumption of a specific serving size were standardized using visual aids to determine a standard unit for portions. Dietary intake assessed using the FFQ was analyzed using nutrition and fitness software (ESHA Food Processor SQL, version 10.1.1, Salem, OR, USA). ESHA was used to estimate a gross mean of daily vitamin D3 intake from food. We also obtained data on current smoking history and physical activity. Individuals were considered to be physically active when they met the target of 150 min of moderate-intensity (or 75 min of vigorous-intensity) per week [29]. 

Solar ultraviolet doses and vitamin D3 production were estimated using the approach described by Godar and colleagues [30]. To do that, we obtained information on the following: Fitzpatrick skin type scale, duration and timing of direct exposure to sunlight per day, the fraction of body exposed, age factor, action spectrum conversion factors (ASCF), and geometric conversion factors (GCF). 

The Fitzpatrick skin type scale is utilized to evaluate the reaction of different types of skin to ultraviolet light [31]. Type I (scores 0–6—pale white) easily burns, does not tan. Type II (scores 7–13—white) typically burns, tans slightly. Type III (scores 14–20—light brown) mild burn, tans consistently. Type IV (scores 21–27—moderate brown) minimally burns, tans. Type V (scores 28–34—dark brown) infrequently burns, tans easily. Type VI (scores 35–36—dark brown or black) never burns. 

Scattered duration and timing of direct exposure to sunlight per day were obtained by asking the participants to estimate the average time spent on outdoor activities with an emphasis on the proportion being exposed to direct sunlight. This was used to calculate Standard Erythemal Dose (SED) [32]. The solar zenith angle was not considered in our research. 

The fraction of body exposed is the body surface exposed to sunlight. The following standard fractions were used: sun on arms and hands only (short-sleeved shirt, head is covered) = 11%; sun on face, neck, arms, and hands (same like before, but no head cover) = 18%; sun on face, neck, arms, hands, and lower legs (wearing shorts and shirt, no head cover) = 32%; sun on the top half of body (stripped up to waist) = 53%; sun on whole body except for one-piece bathing costume (ladies) = 73%; sun on whole body except for swimming costume = 88%; sun on whole body = 100% [30]. 

Age factor encompasses the ability of an adult to synthesize vitamin D3. The ability to produce vitamin D3 is decreased as human age due to decreased 7-dehydrocholesterol in the skin. The following age factor conversion was used: 0–20 years (100% or 1.0), 22–40 years (83% or 0.83), 41–59 years (66% or 0.66), and 60+ years (49% or 0.49) [30,33]. 

The action spectrum conversion factors are the differences between wavelength contributions approximated by the erythemal action spectrum and the previtamin D action spectrum toward previtamin D3 production. ASCFs for Bahrain (26 °N) were compensated with values latitude 30 °N as follows: 1.110 for summer, 1.061 for fall, 0.910 for winter, and 1.065 for spring, respectively [34]. 

The standard vitamin D dose, which represents a horizontal plane or planar doses, is converted to whole-body doses using geometric conversion factors based on a full-cylinder model representing the human body. GCF for Bahrain (26 °N) is 0.580 during the summer and spring and 0.644 during the winter and fall [34]. The daily estimate of synthesized vitamin D3 per day was estimated using the following equations: Estimate vitamin D3 (IU) per day = Vitamin D Dose (VDD) × (4900 IU) × skin type factor × fraction of body exposed × age factor.Standard Vitamin D Dose (SVD) = Standard Erythemal Dose (SED/day) × Action Spectrum Conversion Factor (ASCF).Vitamin D Dose (VDD) = Standard Vitamin D Dose (SVD) × Geometric Conversion Factors (GCF).

To convert vitamin D from IU to mc: 1 IU is approximated to be the biological equivalent of 0.025 mcg cholecalciferol or ergocalciferol [35]. 

### 2.5. Ethical Considerations

This research was approved by the Secondary Healthcare Research Ethics Committee in the Ministry of Health, Bahrain (No.2018/REC/EF023). Before the start of data collection, informed consent was requested and secured from each person included. 

### 2.6. Statistical Analyses

Descriptive statistics were used to a provide summary of the demographic characteristics, health status, and daily vitamin D from diet and sunlight exposure. The arithmetic mean and standard deviation (SD) were utilized for continuous variables, and the count and percentage for categorical variables. A daily dose of vitamin D < 400 IU, serum levels < 30 nmol/L was considered as deficient, levels between 30 nmol/L and 50 nmol/L (≥30, ˂50) were classified as vitamin D insufficiency, and optimal levels were ≥50 nmol/L. Sensitivity, specificity, percentage correctly classified, and receiver operating characteristic (ROC) Area were calculated for the estimated intake of vitamin D using 25(OH)D as reference. Multiple linear regression analysis was performed to assess the association between the dose of vitamin D per day and selected predictors. A statistically significant result was *p*-value < 0.05. All analyses were executed using Stata 16.1 programming [36]. 

## 3. Results

This study involved 192 participants: 96 patients with depression and 96 age- and sex-matched controls. The mean age was approximately 43 years, with 60% being female sex. Table 1 shows the characteristics of the study participants. The results generally show that patients with depression are more likely to be unemployed, single, and overweight or obese. During the study, all patients were on active pharmacological treatments, 42% were on selective serotonin reuptake inhibitors, 35% were on serotonin and norepinephrine reuptake inhibitors, 12% were on tricyclic antidepressants, and the remaining 11% were on others or combined antidepressants therapy.

Table 2 shows the vitamin D status of the study participants. The daily dose of vitamin D is approximately 260 IU (7 mcg) per day for the entire participants (*n* = 192), with 212 IU (5 mcg) per day acquired from sunlight exposure and 84 IU (2 mcg) per day from dietary intake. Only 47 (25%) met the minimum daily recommended dose of ≥400 IU (10 mcg) per day. Patients with depression had a lower intake of vitamin D per day with 234 IU (6 mcg), 153 (4 mcg), and 81 (2 mcg) per day from all sources, sunlight exposure, and dietary intake, respectively. Controls had a higher intake of vitamin D per day with 357 IU (9 mcg), 270 (7 mcg), and 87 (2 mcg) per day from all sources, sunlight exposure, and dietary intake, respectively. Intake of vitamin D from the diet was equal between the two groups *p* = 0.5, but intake from sunlight exposure and cumulative daily intake of vitamin D was statistically significant for the favor of controls *p* = 0.001 and *p* = 0.001, respectively. Serum 25(OH)D for cases with depression and controls were 35 ± 7 nmol/L (ng/mL) and 38 ± 6 nmol/L (ng/mL), respectively. The difference was statistically significant *p* = 0.01. Recent research in Bahrain showed that controls have a mean serum of 39.95 nmol/L [37]. The proportions of persons at the cutoff 25(OH)D ≥ 35 nmol/L were 56% and 76%, and at cutoff 25(OH)D ≥ 40 nmol/L were 21% and 46% for cases and controls, respectively. The difference was significant at both cutoffs points *p* = 0.04 and *p* = 0.01, respectively. See Table 2.

The relationship between serum 25(OD)D and daily vitamin D dose from dietary intake and solar ultraviolet B is presented in Figure 1 and Figure 2, respectively.

The sensitivity, specificity, percentage correctly classified, and ROC Area for the estimated vitamin D against the 25(OH)D as reference were 100%, 79%, 80%, and 89%.

Figure 3 illustrates the intake of vitamin D among patients with depression according to symptoms of severity. Figure 4 illustrates serum 25(OH)D among patients with depression according to symptoms of severity.

One-way analysis of variance (ANOVA) revealed that the mean daily dose of vitamin D for patients with depression did not significantly differ according to symptoms severity as measured by the BDI-II with *p* = 0.15. Patients with mild, moderate, and severe symptoms had a daily dose of 268 IU (7 mcg), 181 IU (5 mcg), and 275 IU (7 mcg) accordingly.

Multiple linear regression analysis showed that the only predictor for vitamin D doses per day is physical activity for both cases with depression and controls *p* = 0.001. Detailed results are presented in Table 3.

## 4. Discussion

To the authors’ best knowledge this is the first study to measure vitamin D doses from solar ultraviolet and dietary intakes in patients with depression. The major finding of this study is that: patients with depression have significantly lower doses of vitamin D compared to age- and sex-matched healthy controls. While dietary intakes of vitamin D are equal in both groups, patients with depression appeared to have statistically significantly less vitamin D from solar ultraviolet B. The proportion of patients with depression meeting the daily recommended dose of vitamin D is less than one out of five. The daily dose of vitamin D did not vary significantly among patients with depression according to symptoms of severity.

A recent laboratory-based study found a very high prevalence of vitamin D deficiency among patients with mental illness with only 18% showing adequate levels of vitamin D [19]. A meta-analysis of fourteen observational studies with approximately 31,500 patients revealed that lower vitamin D levels were found in patients with depression compared to healthy individuals [20]. Our results are consistent with previous research, which suggest that generally 20% of patients with depression have lowered vitamin D and increased vitamin D deficiency.

A low 25(OH)D in depressed patients can be also attributed to antidepressants drug use. Previous research found that antidepressants use, especially tricyclic antidepressants, appeared significantly associated with lower vitamin D [38].

Previous research demonstrated an association between adequate diet and sensible sun exposure to vitamin D deficiency among patients with depression [19]. Our findings suggest that sun exposure plays a more important role in explaining vitamin D deficiency in both patients with depression and healthy controls. It is well documented that low vitamin D can be linked with many health problems including neuropsychiatric disorders [20,39,40,41,42]. Specifically, observational and experimental studies showed a relationship between low levels 25(OH)D and depression [27,42,43,44].

The low doses of vitamin D for solar ultraviolet can be explained by the fact that adults with depression and depressive disorders engage in low levels of physical activity and poor lifestyle behavior [45,46]. Thus, because lower levels of vitamin D may precipitate mental disorders [13,47], a reestablishment of adequate levels may improve mental wellbeing and offer a feasibly adjunct treatment option. This is especially true if it is offered as part of a comprehensive lifestyle intervention that includes an outdoor physical activity with solar light exposure. Recent research showed that vitamin D and exercise have independent desirable influence on mood. Thus, the active engagement in outdoor activities under the sunlight can neutralize the vitamin D deficiency problem and the severity of mood disorders [48]. Sun avoidance inventory (SAI) can be used to examine outlooks towards sun avoidance attitudes in the context of vitamin D deficiency. In our study, we excluded participants who are taking dietary supplements; however, vitamin D exposure through supplementation should be also included in measures of overall vitamin D exposure.

This is the first research to estimate vitamin D doses from solar ultraviolet and dietary intakes in patients with depression using a rigorous approach and using a case-control methodology. Another strength is that we compared the estimated vitamin D doses against serum 25(OH)D. We focused on outpatients with depression to eliminate the role of hospital-based restricted diets and inpatients closed wards policy; however, future studies are needed to compare inpatients vs. outpatients.

## 5. Conclusions

The present study showed that about 80% of patients with depression and 70% of controls do not receive adequate daily doses of vitamin D. Effective detection and interventions on adequate vitamin D levels in patients with depression might prove to be an easy and cost-effective intervention to improve long-term health outcomes.

## Figures and Tables

**Figure 1 nutrients-12-02587-f001:**
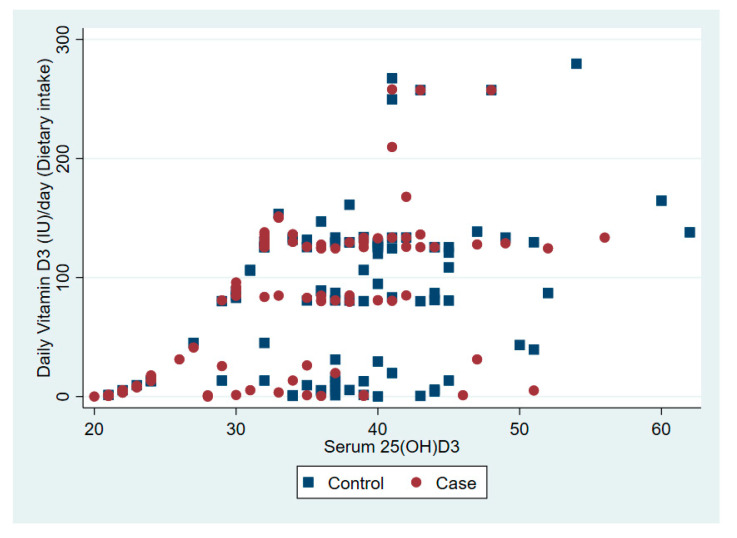
The association between serum vitamin D 25(OH)D and daily vitamin D from dietary intake.

**Figure 2 nutrients-12-02587-f002:**
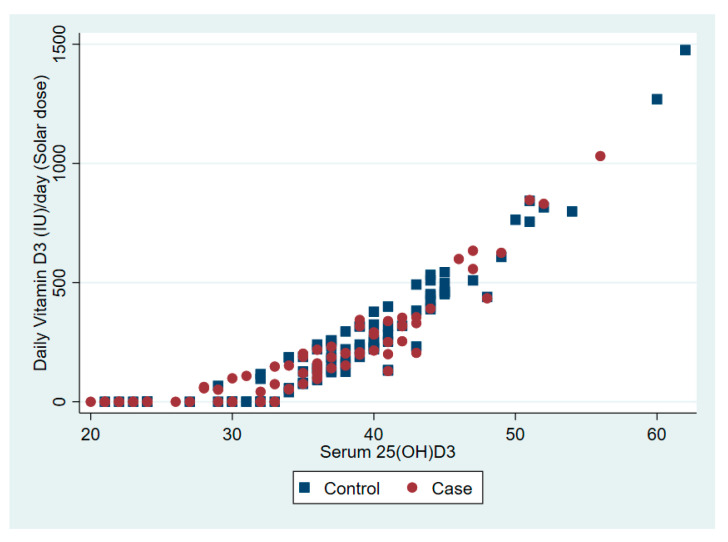
The association between serum vitamin D 25(OH)D and daily vitamin D from solar ultraviolet B.

**Figure 3 nutrients-12-02587-f003:**
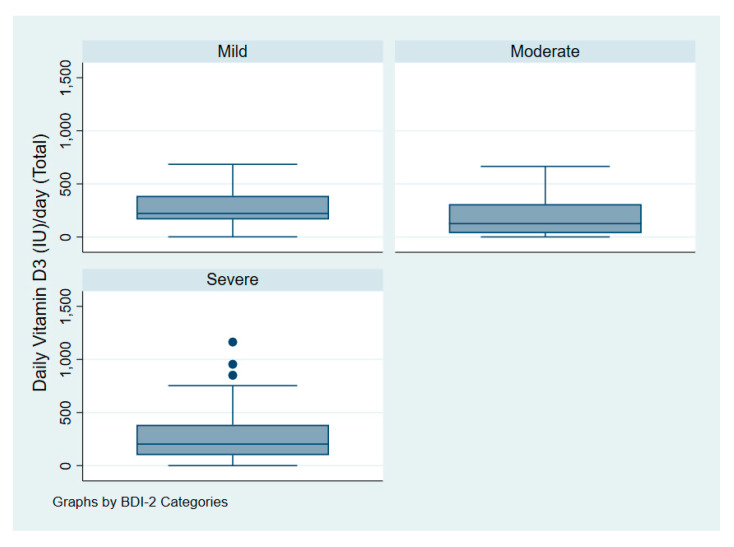
Vitamin D dose (IU/day) of patients with depression according to symptoms severity.

**Figure 4 nutrients-12-02587-f004:**
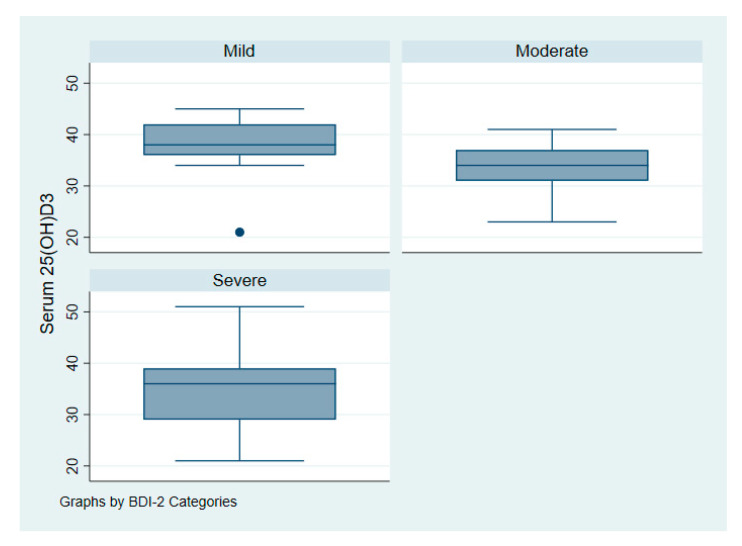
Serum vitamin D 25(OH)D of patients with depression according to symptoms severity.

**Table 1 nutrients-12-02587-t001:** Sociodemographic and anthropometric characteristics of the study participants.

Variable *	Cases, *n* = 96	Controls, *n* = 96	*p*-Value **
Sex			
Male	37 (39%)	37 (39%)	1.0
Female	59 (61%)	59 (61%)	
Job-status			
Employed	27 (28%)	69 (72%)	0.001
Unemployed	69 (72%)	27 (28%)	
Marital status			
Single	48 (50%)	23 (24%)	0.001
Married	48 (50%)	73 (76%)	
BMI classification			
Underweight	4 (4%)	2 (2%)	0.25
Normal	26 (27%)	34(35%)	
Overweight	30 (31%)	35 (37%)	
Obese	36 (38%)	25 (26%)	
Current tobacco smoker	37%	10%	0.001
Beck Depression Inventory-II (BDI-II)			
Mild	13 (13%)	Not applicable	-
Moderate	40 (42%)		
Severe	43 (45%)		
Age (year)	44 ± 13	43 ± 15	0.4
Weight (kg)	76 ± 19	75 ± 17	0.63
Height (cm)	163 ± 10	165 ± 10	0.13
BMI (kg/m^2^)	29 ± 7	28 ± 6	0.11
Body fat percentage (%)	35 ± 12	33 ± 10	0.09
Total body water percentage (%)	36 ± 6	36 ± 7	0.91
Body surface area (m^2^)	2 ± 0.2	2 ± 0.2	0.98
Lean mass (kg)	49 ± 8	49 ± 8	0.68
Fat mass (kg)	28 ± 13	26 ± 10	0.31
Serum 25(OH)D (nmol/L) *** ^a,b^	35 ± 7	38 ± 6	0.01

* Frequency count and (%) OR Mean ± SD; ** Independent samples *t*-test or Pearson’s Chi-Squared; *** ^a^ To convert to ng/mL, divide by 2.5, ^b^ data available for 43 cases and 50 controls.

**Table 2 nutrients-12-02587-t002:** Vitamin D status of the study participants.

* Variable	Cases, *n* = 96				Controls, *n* = 96				*p*-Value **
	Mean	SD	SE	95%CI	Mean	SD	SE	95%CI	
Vitamin D intake from diet per day (IU)	81	65	7	68–94	87	66	7	74–101	0.50
Vitamin D synthesis from sunlight per day (IU)	153	206	21	111–195	270	260	27	218–323	0.001
Vitamin D per day (IU)	234	275	23	189–280	357	275	28	301–413	0.001
Share of Vitamin D from diet per day	35%				25%				0.11
Share of Vitamin D from sunlight exposure per day	65%				75%				0.11
Compliance with the recommended minimum daily intake (400 IU per day)	18 (19%)				29 (30%)				0.048
Vitamin D according to 25(OH)D Optimal—≥50 nmol/L Insufficient—≥30 ˂50 nmol/L Deficient—<30 nmol/L	1 (2%) 34 (79%) 8 (19)				3 (6%) 44 (88%) 3 (6%)				0.13
Serum 25(OH)D ≥ 30 nmol/L	35 (83%)				47 (94%)				0.06
Serum 25(OH)D ≥ 35 nmol/L	24 (56%)				38 (76%)				0.04
Serum 25(OH)D ≥ 40 nmol/L	9 (21%)				23 (46%)				0.01
Serum 25(OH)D ≥ 45 nmol/L	3 (7%)				4 (8%)				0.90
Serum 25(OH)D ≥ 50 nmol/L	1 (2%)				3 (6)				0.40

* Frequency count and (%) OR Mean ± SD; ** Independent samples *t*-test or Pearson’s Chi-Squared.

**Table 3 nutrients-12-02587-t003:** Association * between total vitamin D doses and selected predictors.

**Cases with depression (*n* = 96)**		
	**Outcome variable: Daily vitamin D Dose**	
**Explanatory Variables**	**β**	***p*-Value**
Education level	60	0.12
Smoking	1	1
Physical activity	318	0.001 *
**Controls (*n* = 96)**		
	**Outcome variable: Daily vitamin D Dose**	
**Explanatory Variables**	**β**	***p*-Value**
Education level	111	0.08
Smoking	−52	0.60
Physical activity	267	0.001 *

* Multiple linear regression analysis—Adjusting for age, sex, caloric intake, social status, and job status.

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
