# Peer review of "Vitamin D Doses from Solar Ultraviolet and Dietary Intakes in Patients with Depression: Results of a Case-Control Study"

_nutrients, 2020, doi:10.3390/nu12092587_

Round 1

Reviewer 1 Report

Appropriate study design, novel measures, strong manuscript

Hypothesis: Healthy controls are exposed to higher daily vitamin D3 doses from dietary and sunlight exposure compared to cases with depression. Also, they hypothesize the severity of depressive symptoms is also associated with level of exposure to vitamin D3.

I think this study is relevant and interesting because it is imperative to understand the relationship between types of D3 exposure and depression. This study highlighted lack of D3 exposure for both cases and controls and the need to meet the recommended dose of D3.

There are a few studies studying supplementation and depression, as well as overall sunlight exposure and depression. However, I could not find a study in PubMed that looked at D3 exposure from both diet and sunlight and its relation to depression.

Using a D3 exposure that includes diet and sunlight exposure adds to the subject area. However, in future study designs, I think that D3 exposure through supplementation should also be included in measures of overall D3 exposure. 

The paper was well written, methods and results were clear and easy to read. Are the conclusions consistent with the evidence and arguments presented? Do they address the main question posed? The conclusions are consistent with the evidence and arguments presented

Author Response

Dear Editors and Reviewers, we are grateful for your consideration of this manuscript, and we also very much appreciate your time and suggestions, which have been helpful in improving the manuscript.

All the comments we received on this study have been taken into account when preparing the revision. We have addressed each and every comment.

We hope that these changes to the manuscript will facilitate the decision to be made by the journal.

Sincerely,

The authors

Appropriate study design, novel measures, strong manuscript

Thank you for this nice comment.

Hypothesis: Healthy controls are exposed to higher daily vitamin D3 doses from dietary and sunlight exposure compared to cases with depression. Also, they hypothesize the severity of depressive symptoms is also associated with level of exposure to vitamin D3.

Perfectly correct.

I think this study is relevant and interesting because it is imperative to understand the relationship between types of D3 exposure and depression. This study highlighted lack of D3 exposure for both cases and controls and the need to meet the recommended dose of D3.

There are a few studies studying supplementation and depression, as well as overall sunlight exposure and depression. However, I could not find a study in PubMed that looked at D3 exposure from both diet and sunlight and its relation to depression.

Thank you, true, the novelty of this study is including both dietary and UVB (solar dose) of vitamin D.

Using a D3 exposure that includes diet and sunlight exposure adds to the subject area. However, in future study designs, I think that D3 exposure through supplementation should also be included in measures of overall D3 exposure. 

Thanks, we agree and we added that in our recommendations.

We added the following: “In our study we excluded participants who are taking dietary supplements; however, vitamin D exposure through supplementation should be also included measures of overall vitamin D exposure.“

The paper was well written, methods and results were clear and easy to read. Are the conclusions consistent with the evidence and arguments presented? Do they address the main question posed? The conclusions are consistent with the evidence and arguments presented.

Thank you for this nice comment and encouragement.

Reviewer 2 Report

This manuscript demonstrated in a quite huge number of individuals (cases-controls) a link between low levels of solar ultraviolet vitamin D3 and depression.

The work sounds interesting and the experimental plan is well designed since it took into account many variables that could have interfered with the results related to vitamin D3 dosages.

I found incorrect the representation of the results in Figures 1, 2, 3, 4. To appreciate differences or not of vitamin D3 dosages, the groups considered should be on the same graph.

I strongly suggest changing the representation of the figures.

Author Response

Dear Editors and Reviewers, we are grateful for your consideration of this manuscript, and we also very much appreciate your time and suggestions, which have been helpful in improving the manuscript.

All the comments we received on this study have been taken into account when preparing the revision. We have addressed each and every comment.

We hope that these changes to the manuscript will facilitate the decision to be made by the journal.

Sincerely,

The authors

This manuscript demonstrated in a quite huge number of individuals (cases-controls) a link between low levels of solar ultraviolet vitamin D3 and depression.

Thank you very much for these nice comments.

The work sounds interesting and the experimental plan is well designed since it took into account many variables that could have interfered with the results related to vitamin D3 dosages.

Thank you very much for these nice comments.

I found incorrect the representation of the results in Figures 1, 2, 3, 4. To appreciate differences or not of vitamin D3 dosages, the groups considered should be on the same graph. I strongly suggest changing the representation of the figures.

Figure 1 and Figure 2 were changed to include both groups in the same graph. We think it look more appealing and deliver the picture very well.

Figure 3 and Figure 4 are only patients by severity, so no need to change.
